# 2-Methoxy-4-Vinylphenol as a Biobased Monomer Precursor for Thermoplastics and Thermoset Polymers

**DOI:** 10.3390/polym15092168

**Published:** 2023-05-02

**Authors:** Alexandros E. Alexakis, Thayanithi Ayyachi, Maryam Mousa, Peter Olsén, Eva Malmström

**Affiliations:** 1Division of Coating Technology, Department of Fibre and Polymer Technology, School of Engineering Sciences in Chemistry, Biotechnology and Health, KTH Royal Institute of Technology, Teknikringen 56-58, SE-100 44 Stockholm, Sweden; 2Wallenberg Wood Science Center, Department of Fibre and Polymer Technology, School of Engineering Sciences in Chemistry, Biotechnology and Health, KTH Royal Institute of Technology, Teknikringen 56, SE-100 44 Stockholm, Sweden; 3Division of Biocomposites, Department of Fibre and Polymer Technology, School of Engineering Sciences in Chemistry, Biotechnology and Health, KTH Royal Institute of Technology, Teknikringen 56-58, SE-100 44 Stockholm, Sweden

**Keywords:** lignin, biomass, emulsion, crosslinking, curing, thiol-ene

## Abstract

To address the increasing demand for biobased materials, lignin-derived ferulic acid (FA) is a promising candidate. In this study, an FA-derived styrene-like monomer, referred to as 2-methoxy-4-vinylphenol (MVP), was used as the platform to prepare functional monomers for radical polymerizations. Hydrophobic biobased monomers derived from MVP were polymerized via solution and emulsion polymerization resulting in homo- and copolymers with a wide range of thermal properties, thus showcasing their potential in thermoplastic applications. Moreover, divinylbenzene (DVB)-like monomers were prepared from MVP by varying the aliphatic chain length between the MVP units. These biobased monomers were thermally crosslinked with thiol-bearing reagents to produce thermosets with different crosslinking densities in order to demonstrate their thermosetting applications. The results of this study expand the scope of MVP-derived monomers that can be used in free-radical polymerizations toward the preparation of new biobased and functional materials from lignin.

## 1. Introduction

Modern lifestyle is rife with thermoplastics and thermosets, due to their low cost, ease of manufacture, property profile, convenience of use, durability, and abundance of platform monomers derived from petroleum sources [1]. Petroleum-based platform chemicals are synthesized from olefinic or aromatic building blocks to yield plastics. During the last few decades, it has been widely recognized that CO_2_ (g) emissions lead to human-induced climate change [2], which has resulted in an increasing call for reduced dependence on petroleum-based resources [3]. However, only a tiny amount of the annual production of plastics originates from biobased resources [4]. In addition, the amount of identified petroleum reserves is not sufficient to support humankind consumption for more than approximately five decades [5]. On the other hand, a variety of non-fossil feedstocks from plants are renewable and could be produced sustainably over the long term. Therefore, the transition from fossil-based to a biobased economy is not only warranted, but also provides many opportunities [6,7,8].

Even though biobased materials derived from renewable resources pose a great interest, their upscale production in an ecofriendly manner is still challenging. Moreover, the targeted end-application governs the preferred chemical composition of the biobased candidate, thus restricting possible alternatives. Although monomers suitable for step-growth polymerization are readily available from biobased sources, monomers for chain growth polymerization, which require an accessible double bond, are less explored [9]. It is highly motivated to scout biobased resources that can provide monomers containing double bonds since 50% of all polymers are based on chain-growth polymerization [10,11,12]. Moreover, the desired chemistries are not easily accessible; hence, research efforts have focused on converting those feedstocks into platform compounds similar to those obtained from petroleum sources through different processes (e.g., chemical or enzymatic) [9,13,14].

Within the scope of biobased alternatives, biomass is an abundantly available raw material in many parts of the world. Wood primarily comprises cellulose, with the remainder being hemicellulose and lignin, the latter of which varies between 15% and 30% depending on the wood species [15]. Lignin is an aromatic polymeric network built up from three different monolignol units in varying ratios, depending on the species [16]. One such monolignol is coniferyl alcohol, and it is the most abundant monolignol comprising the lignin network in softwoods [17]. The oxidized form of coniferyl alcohol is denoted as ferulic acid (FA, Figure 1), which, in addition to being found in trees and grasses, can be found in edible produce, such as grains and fruits [18]; it can also act as a precursor for vanillin [19]. Typically, it is isolated via alkaline or acidic extraction methods [20,21]; more recently, enzyme-assisted extractions have also been reported, which could enhance the production of FA from biomass [22]. Additionally, it has been shown that FA can be produced through the transformation of eugenol, which is another lignin-derived compound [23]. It is estimated that the global FA market size will reach 135 million USD in the next decade, expecting to exceed 750 tons, with an annual growth rate of 7.9% [24]. Hitherto, FA has been used industrially in the medical and food sector due to its antioxidative properties [25], as well as in research in polycondensation reactions to form polyesters or polyamides by taking advantage of its pristine or modified hydroxyl and carboxylic acid functionalities [26,27,28]. It has been shown that FA can be readily decarboxylated to form the styrene-like monomer 2-methoxy-4-vinylphenol (MVP, Figure 1) [29,30]. Recently, MVP was successfully polymerized via ionic polymerization with and without the modification of its phenolic group with silyl ethers [31,32]. Moreover, silyl-protected and epoxidized MVP were also investigated in controlled radical polymerizations, yielding polymers with a wide range of thermal and chemical properties [33,34].

Although FA-derived monomers have shown properties that could be well suited to replace fossil-based monomers in polymerizations, their modifications and suggested end-applications have so far been limited [31,32,33,34,35]. In this work, we expand the scope and, therefore, emphasize the potential of FA as a platform monomer by creating a library of biobased monomers and exploring their structure–property relationship in thermoplastic and thermosetting applications. Initially, MVP was produced via the decarboxylation of FA, whereafter its phenolic group was modified to yield monomers that can undergo solution and emulsion polymerization, toward thermoplastic applications. Moreover, MVP was modified to synthesize divinylbenzene (DVB)-like monomers of different aliphatic chain lengths, which were thermally cured with thiol crosslinkers to produce thermosetting films. The resulting materials were characterized by means of chemical (nuclear magnetic resonance (^1^H-, ^13^C-, and 2D-NMR), Fourier-transform infrared spectroscopy (FT-IR), polarity (contact angle), and chemical stability tests), thermal (differential scanning calorimetry (DSC) and thermogravimetric analysis (TGA)), morphological (dynamic light scattering (DLS)), and mechanical (dynamic mechanical analysis (DMA)) techniques, which provided a holistic view of their final properties.

## 2. Materials and Methods

A list of all materials used in this study can be found in the Appendix A. Furthermore, the synthetic protocols for 2-methoxy-4-vinylphenol (MVP), methyl 2-(2-methoxy-4-vinylphenoxy)acetate (MVP-AcOMe), 2-(2-methoxy-4-vinylphenoxy)acetic acid (MVP-AcOH), and divinylbenzene (DVB)-like monomers based on MVP, including the instrumentation used for their characterization, can be found in the Appendix A.

### 2.1. 1-(Benzyloxy)-2-Methoxy-4-Vinylbenzene (MVP-Bz)

MVP (5.0 g, 33 mmol, 1 equiv.) was mixed with acetone (85 mL) in a 250 mL round-bottom flask under stirring, followed by the addition of K_2_CO_3_ (5.1 g, 36 mmol, 1.1 equiv.) and, subsequently, the catalyst 18-crown-6 (0.5 g, 1.7 mmol, 0.05 equiv.). Benzyl bromide (BzBr) (6.3 g, 37 mmol, 1.1 equiv.) was added dropwise, and the flask was transferred to an oil bath and left to react under reflux (56 °C) overnight. Then, the product was concentrated, and Et_2_O (25 mL) was added. The organic phase was washed with water (3 × 8 mL), dried over MgSO_4_, and concentrated in rotavap to obtain MVP-Bz as a white solid (yield 90%). A final recrystallization step in absolute ethanol was necessary. ^1^H-NMR (400 MHz, DMSO-*d_6_*, Appendix A) δ 7.48–7.28 (m, 5H), 7.11 (d, J = 1.9 Hz, 1H), 7.00 (d, J = 8.3 Hz, 1H), 6.95 (dd, J = 8.3, 2.0 Hz, 1H), 6.64 (dd, J = 17.6, 10.9 Hz, 1H), 5.71 (dd, J = 17.6, 1.1 Hz, 1H), 5.13 (dd, J = 10.9, 1.0 Hz, 1H), 5.08 (s, 2H), 3.80 (s, 3H). ^13^C-NMR (101 MHz, DMSO-*d_6_*, Appendix A) δ 149.24, 147.77, 137.08, 136.39, 130.53, 128.35, 127.78, 127.72, 119.16, 113.51, 112.14, 109.39, 69.89, 55.54.

### 2.2. Solution Polymers P(MVP-AcOMe), P(MVP-AcOH), and P(MVP-Bz)

The protocol for the solution polymerization of the MVP-derived monomers was identical for all monomers; hence, only the case of MVP-AcOMe is described below. MVP-AcOMe (1.0 g) was dissolved in toluene (3 mL) in a 10 mL round-bottom flask under stirring. Then, 2,2′-azobisisobutyronitrile (AIBN) (0.01 g, 1 wt.% of MVP-AcOMe) was added to the flask. The reaction mixture was kept under argon for 20 min, after which the reaction flask was immersed in an oil bath preheated to 70 °C, thus initiating the polymerization. Aliquots were drawn at different time intervals to monitor the kinetics of the polymerization by ^1^H-NMR. The polymerization was stopped when high conversion was reached, i.e., after 24 h for P(MVP-AcOH) and 25 h for P(MVP-AcOMe) and P(MVP-Bz), by cooling the flask at ambient temperature and exposing the reaction mixture to air. Finally, the crude polymer was precipitated in MeOH, filtered, washed with excess of MeOH, and dried under vacuum overnight.

In the case of P(MVP-AcOH), acetonitrile was used as a solvent instead of toluene, and MeOH was replaced by toluene during the precipitation step. The kinetics of the polymerization for all investigated monomers, followed by ^1^H-NMR, can be found in Appendix A, and the monomer conversion is shown in Appendix A.

### 2.3. Emulsion Copolymers

The MVP-derived monomers were copolymerized with styrene in emulsion polymerization. Below, the case of P(S_90_-MVP-AcOMe_10_) is described, wherein the number in subscript denotes the targeted weight percentage of the monomer in the final polymer. Typically, sodium dodecyl sulfate (SDS) (0.2 g, 4 wt.% of total monomers) and potassium persulfate (KPS) (0.05 g, 1 wt.% of total monomers) were dissolved in a two-neck round-bottom flask in water (20 mL) (targeting 20 wt.% of total dry content, assuming 100% conversion). Then, styrene (4.5 g, 43 mmol) was added and left to stir for 5 min before adding MVP-AcOMe (0.5 g, 4.7 mmol). The reaction flask was equipped with a condenser and a magnetic stirrer, and then purged with argon for 15 min before immersing the flask in a preheated oil bath set at 80 °C at 500 rpm. The reaction proceeded for 3 h to reach conversions higher than 80%. The emulsion was coagulated using 5 wt.% of alum solution (AlK(SO_4_)_2_·12H_2_O) under intense stirring, and the polymer was dried in vacuum overnight. The final conversion was estimated gravimetrically (Equation (S2)). In total, three emulsions were prepared, denoted P(S_90_-MVP-AcOMe_10_), P(S_90_-MVP-Bz_10_), and P(S_90_-MVP-AcOMe_5_-MVP-AcOH_5_), as well as a polystyrene (PS) reference.

### 2.4. Preparation of Thiol-Ene Thermoset Films

The synthesized crosslinkers were cured thermally with thiols to obtain thiol-ene thermosets; a combination of 1,3-MVP, 1,6-MVP, and 1,10-MVP with two different thiol-bearing reagents, trimethylolpropane tris(3-mercaptopropionate) (3T) or pentaerythritol tetrakis(3-mercaptopropionate) (4T), resulted in six different films. In each combination, a 1:1 ratio between the thiol and -ene functional groups was maintained. Briefly, the case of 1,3-MVP with 3-thiol, referred to as X(1,3-MVP–3T), is described. First, 1,3-MVP (0.14 g, 0.39 mmol, 1 equiv.) was transferred to a Teflon mold and melted in the oven at 125 °C for 30 min. Then, 3T (0.11 g, 0.26 mmol, 0.67 equiv.) was added and mixed by stirring with a pipette. The resins were left in the oven at 125 °C for 24 h. Thereafter, the mold was cooled to room temperature, and the cured thermoset films were removed and stored in a glass vial until further characterization. The amounts for the preparation of the remaining thermoset films can be found in Appendix A.

When 4T was used, the procedure remained the same but the amount added was 0.1 g (0.2 mmol, 0.5 equiv.).

### 2.5. Chemical Stability Test

The chemical stability of the prepared thermosetting films was tested by immersing the films (approximately 10 mg) into vials containing 10 mL of different aqueous or organic solutions. In this study, NaOH (1 M), HCl (1 M), DMSO, CHCl_3_, and acetone were used. The samples were left in the aforementioned solvents on a shaking table for 7 days and, thereafter, evaluated visually.

## 3. Results

This study focuses on expanding the scope of biobased platform monomers derived from ferulic acid (FA) in radical polymerization by demonstrating their versatility in thermoplastic and thermosetting applications. The study is divided into two parts: (i) synthesis of biobased monomers derived from FA and their homo- and copolymerization via solution and emulsion polymerization techniques, respectively; (ii) synthesis of biobased divinylbenzene (DVB)-like monomers and their use in thermally cured thermosetting films (Figure 1).

### 3.1. Thermoplastic Applications

It has been shown that the styrene-like monomer 2-methoxy-4-vinylphenol (MVP) can be derived from FA through decarboxylation [29,30]. However, little work has been reported in the literature using MVP as a precursor to new biobased functional monomers that can undergo radical polymerization [31,34].

MVP contains a terminal double bond and a phenol, which enable further functionalization (Figure 1). Both functional groups are highly reactive, which could lead to unfavorable autopolymerization, e.g., due to the radical scavenging effect, as shown for other phenolics [36].

Withstanding the fact that the double bond should remain intact to subsequently participate in free-radical polymerization, the phenolic group was functionalized to prepare biobased derivatives. Due to its highly reactive character, MVP was functionalized within the timeframe it was considered stable (Appendix A).

The first MVP derivative was 2-(2-methoxy-4-vinylphenoxy)acetate (MVP-AcOMe, Appendix A), an aliphatic modification resulting from the reaction between MVP and methyl bromoacetate (MeBrAc, Figure 1). The second derivative was 1-(benzyloxy)-2-methoxy-4-vinylbenzene (MVP-Bz, Appendix A), an aromatic modification accomplished by the reaction with benzyl bromide (BzBr, Figure 1). In addition to these hydrophobic derivatives, a hydrophilic modification of MVP, referred to as 2-(2-methoxy-4-vinylphenoxy)acetic acid (MVP-AcOH, Appendix A), was prepared by using MVP-AcOMe as the intermediate (Figure 1). The yield for all reactions was higher than 70%, suggesting that MVP can act as a platform to yield polymerizable derivatives with hydrophobic and hydrophilic characteristics. Moreover, their chemical composition is in line with other lignin monomers and C–O-linked fossil-based commercial polymers, making them applicable for replacing fossil-based monomers (Appendix A) [37].

Although the monomers obtained from the MVP platform were derived from natural resources, the use of halogenated reagents employed in their synthetic routes in this study was hypothesized to impact their overall ecofriendly character. Hence, atom efficiency (AE) [38], environmental factor (E) [39,40], and biobased carbon content (BBCC) [41] (Appendix A) were considered. AE is a measure of the efficiency of a reaction, expressed as a percentage. Hence, a higher AE percentage denotes greater efficiency [38]. The AEs for MVP, MVP-AcOMe, and MVP-Bz (~75%) were greater than that for MVP-AcOH (64%), because the latter is the result of a two-step process, with the MVP-AcOMe being the intermediate. Additionally, E expresses the environmental impact of a synthetic process by determining the waste generated [39,40]. The ideal value for E is 0 (no waste); a higher value of E denotes a more negative environmental impact of a process. The E-values for the synthetic approaches used (~19) were within range of other processes involving lignin-derived monomers (Appendix A) [42] except for MVP-AcOH (~135), due to the two-step procedure and number of solvents used. Moreover, a closer value of BBCC to 100% denotes a greater amount of biobased carbon in the final product [41]. For all monomers, except for MVP-Bz, the BBCC was higher than 75%, which is higher than other eugenol-derived functional monomers (Appendix A) [41]. This is because the substituting group used in the case of MVP-Bz had the same number of carbons as MVP, thus reducing the overall BBCC.

The biobased monomers were initially polymerized via solution polymerization to prepare the appropriate homopolymers. The monomer conversion was followed by ^1^H-NMR, and the kinetics are shown in Appendix A. In Appendix A, it can be seen that all biobased homopolymers exhibited similar apparent polymerization rates; however, MVP-Bz exhibited the lowest conversion rate among the investigated monomers, i.e., 75% compared to 90%. This may have been due to the bulkiness of the modifying side-group, which would have enhanced the steric hindrance during polymerization of the final polymer [43].

The molecular weight (M_n_) of the homopolymers was characterized by size-exclusion chromatography (SEC) with dimethyl formamide (DMF) as the eluent (see Table 1). The thermal properties of the precipitated homopolymers were investigated by DSC and TGA (Table 1, Figure 1a). All homopolymers exhibited glass transition temperatures (*T*_g_ s) lower than polystyrene (PS). This may have been due to the increased mobility of the substituting group, which would have increased the free volume [44]. Additionally, the thermal degradation of the homopolymers under nitrogen atmosphere, expressed by *T*_deg_, showed a fast degradation in the case of MVP-AcOH (around 100 °C), which could be attributed to the cleavage of the –COOH group (corresponding to approximately 20 wt.%), as shown in the literature for the case of methacrylic acid (MAA) (Figure 1a and Appendix A) [45].

To demonstrate the radical polymerization of the MVP-platform monomers in water and reduce the fossil-based content in commercial plastics, without compromising their properties, the MVP-derived monomers were copolymerized with styrene via emulsion polymerization. It was hypothesized that MVP, due to its unprotected phenol, would act as a radical scavenger during radical polymerization; thus, only the protected MVP monomers were considered [36]. It was found that, when the biobased content exceeded 10 wt.%, the resulting emulsions were unstable. Furthermore, MVP-AcOMe and MVP-Bz were miscible with styrene, whereas MVP-AcOH could dissolve in neither styrene nor water, despite the presence of a carboxylic acid, which could be attributed to the pH being slightly acidic (pH 4). Thus, a third comonomer had to be used, i.e., MVP-AcOMe.

Each copolymer was named according to the targeted styrene and MVP-AcOMe, MVP-AcOH, or MVP-Bz content in subscript. After 3 h of polymerization, emulsions with high monomer conversions were obtained (Table 1 and Appendix A). The hydrodynamic diameters (D_H_) recorded by dynamic light scattering (DLS) were similar for all samples except for P(S_90_-MVP-AcOMe_5_-MVP-AcOH_5_), which was higher (Appendix A). This can be explained when considering the polydispersity index (PdI) and the lower monomer conversion achieved compared to the remaining emulsions. It was observed that both P(S_90_-MVP-AcOMe_10_) and P(S_90_-MVP-Bz_10_) were monodispersed, whereas P(S_90_-MVP-AcOMe_5_-MVP-AcOH_5_) showed a range of different sizes (Appendix A). This may have been due to the poor stability and complexity of the system, being the result of three different comonomers with competing reactivities.

The thermal properties of the copolymers are summarized in Table 1. It can be seen that the emulsion copolymers exhibited *T*_g_ values very close to PS, as also verified by the calculated theoretical *T*_g_ (Equation (S3)). This is because PS comprised the majority of the composition (Appendix A). Additionally, the degradation patterns of all emulsion copolymers exhibited similar trends to PS with slightly lower *T*_deg_. According to the literature, PS degrades at around 375 °C which is similar to the major degradation observed in Figure 2 [46]. Moreover, an initial mass loss was observed around 100 °C, possibly due to some residual styrene still being present in the dried polymer (Appendix A).

To summarize this part of the work, we propose simple synthetic approaches to produce biobased derivatives of FA that can later be polymerized using different techniques toward thermoplastic applications. This was exemplified by the synthesis of homo- and copolymers of the biobased monomers in solution and emulsion polymerization. The thermoplastics exhibited good emulsion properties without compromising the thermal and morphological profile of PS, thus demonstrating their potential in partially replacing PS for future coating applications.

### 3.2. Thermosetting Applications

Taking advantage of the styrene-like functionality of MVP, biobased difunctional monomers were prepared by reacting the phenol with brominated alkanes, resulting in divinylbenzene (DVB)-like monomers (Figure 1). In detail, two MVP units were linked with an aliphatic chain consisting of three, six, or 10 carbons, referred to as 1,3-MVP, 1,6-MVP, or 1,10-MVP, respectively. The increase in aliphatic chain length was hypothesized to reduce the *T*_g_ values of the networks when crosslinked with thiol-bearing reagents and, by extension, to modulate the mechanical properties [44,47]. 

The protocol used in this study resulted in relatively high yields with satisfactory biobased characteristics, as described through AE, E, and BBCC (Appendix A). AE reached high values, i.e., on average 70%, for all DVB-like monomers synthesized. E ranged between 60 and 73, which is higher than the monomers synthesized for the thermoplastic applications. This is because the synthetic approach used to prepare the DVB-like monomers required additional purification steps including additional solvents [42]. The BBCC was found to decrease from 85% to 65% upon increasing the aliphatic chain length connecting the MVP units, which resulted in similar BBCC values to other functional biobased monomers in the literature (Appendix A) [41].

Two thiol-bearing crosslinkers were used, containing three or four thiols on average, referred to as 3T and 4T, respectively, to investigate the effect of both the aliphatic chain length and the number of crosslinks on the thermosetting films. The protocol to prepare the thermosets was based on the melting temperature of the DVB-like monomers (around 100 °C, Appendix A). An initial investigation of the crosslinking reaction was carried out in DSC by mixing the thiol-bearing reagents with the DVB-like monomers and monitoring the curing procedure (Appendix A). According to the curing peaks listed in Appendix A, the lowest curing temperature was ~110 °C. However, the DSC experiments were performed under nitrogen atmosphere, which was hypothesized to reduce the curing temperature. Therefore, higher temperature was used for the preparation of thermosets. In particular, the thermosets were prepared in an oven by initially melting the DVB-like monomers at 125 °C, mixing them with the thiol crosslinkers, and then leaving them for 24 h at 125 °C (Appendix A). The extent of curing was assessed by the almost complete disappearance of C=C (~1650 cm^−1^) and –SH (~2600 cm^−1^) peaks in FT-IR spectra (Appendix A), indicating a high curing degree [48].

This protocol was chosen for two reasons: (i) the reduction in undesired chemicals such as solvents and initiators; (ii) the relatively low melting point of the DVB-like monomers, which acted as solvents for the thiol-bearing crosslinkers.

The thermal properties of the thermosets were primarily affected by two parameters: (i) the aliphatic chain length between the DVB-like monomers; (ii) the number of crosslinks. As seen in Table 2 and Appendix A, by increasing the aliphatic chain length, the *T_g_* and *T_m_* values decreased as shown before for the MVP-derived homopolymers, due to the increase in chain mobility [44]. Moreover, *T*_g_ was increased by increasing the number of crosslinks between the DVB-like monomers and the thiol crosslinkers, since the mobility was decreased [47]. Furthermore, according to the TGA results (Appendix A), *T*_deg_ was seemingly unaffected by the aliphatic length between the MVP units when the standard deviation was accounted for. However, *T*_deg_ seemingly increased upon increasing the number of crosslinks when comparing the thermosets that were cured with 3T or 4T. This is to be expected, as increasing the number of crosslinks should improve the thermal properties [49].

The thermomechanical properties of the resulting films were investigated using DMA, and the storage modulus (E′) and tanδ were plotted against temperature (Figure 2).

As shown in Figure 2a, the DMA thermograms indicated a typical thermoset behavior characterized by a glassy plateau and a rubbery plateau below and above the glass transition, respectively. Initially, the samples were equilibrated at −50 °C, below *T*_g_. In that glassy region, the storage modulus ranged between 1.4 and 2.4 GPa (Table 2). The storage modulus values are similar to those of other thermosets containing small amounts of DVB used as the crosslinker [50], as well as sobrerol diacrylate thermosets prepared via thiol-ene chemistry [51]. However, these values did not indicate a clear trend that could be related to the aliphatic chain length or the crosslinking density (v_e_). By increasing the temperature, a sharp transition was observed, attributed to the glass transition region. The *T*_g_^tanδ^ was also evaluated from the peak of tanδ (Figure 2b, Table 2), which was found to be higher than that obtained from DSC for all samples, as also reported elsewhere [48,52]. However, X(1,6-MVP–3T) and X(1,6-MVP–4T) exhibited the highest *T*_g_^tanδ^, deviating from the trend previously observed by DSC. This may have been due to homopolymerization of the DVB-like monomers during the initial melting step in the oven, as described previously from the DSC results (Appendix A) [51]. The degree of homopolymerization was indirectly determined through the gel content (Table 2). Only thermosets X(1,10-MVP–3T), X(1,10-MVP–4T), and X(1,3-MVP–3T) exhibited a high gel content, i.e., >90%. Although this could have been avoided by preparing larger mixing batches in the presence of solvent [53], as described previously, in this work, we aimed for solvent-free procedures, thus reducing undesired chemical waste.

**Table 2 polymers-15-02168-t002:** Properties of MVP-based crosslinked thermosets.

Sample Name	$E_{-50\;{}^{\circ}C}^{\prime }$ (MPa) ^1^	$E_{80\;{}^{\circ}C}^{\prime }$ (MPa) ^2^	$T_{g}^{tan\delta }$ (°C) ^3^	$T_{g}$ (°C) ^4^	$v_{e}$ (mol/m^3^) ^5^	Gel Content (%) ^6^
X(1,3-MVP–3T)	1350 ± 300	2.4 ± 0.8	31 ± 2	29 ± 2	0.3 ± 0.1	98
X(1,3-MVP–4T)	2000 ± 560	7.5 ± 1.1	38 ± 1	34 ± 1	0.9 ± 0.2	67
X(1,6-MVP–3T)	2370 ± 40	2.3 ± 0.3	40 ± 1	22 ± 2	0.3 ± 0.1	81
X(1,6-MVP–4T)	2030 ± 290	6.3 ± 0.3	53 ± 2	36 ± 2	0.7 ± 0.1	78
X(1,10-MVP–3T)	2120 ± 230	2.8 ± 0.8	30 ± 1	4.5 ± 0.1	0.3 ± 0.1	94
X(1,10-MVP–4T)	2040 ± 75	7.0 ± 1.2	33 ± 1	18.9 ± 0.4	0.8 ± 0.2	95

^1^ Storage modulus at −50 °C. ^2^ Storage modulus at 80 °C. ^3^ Obtained from the maximum of tanδ. ^4^ Obtained from DSC. ^5^ Crosslinking density evaluated using Equation (S4) [54]. ^6^ Gel content calculated using Equation (S5) [51]. All thermomechanical data are average values over three samples.

For the case of 3T crosslinked samples, the thermosets broke at the grips of the DMA instrument after reaching 80 °C. In addition, these samples exhibited lower storage moduli compared to their 4T counterparts. Moreover, the v_e_ estimated at 80 °C according to Equation (S4) was calculated to be three times higher for 4T compared to 3T, which was expected as more –SH units were available for crosslinking in the case of 4T compared to 3T, thus increasing the v_e_. This was qualitatively observed by the rightward shift of the storage moduli in the glass transition region [52]. 

To evaluate the robustness of the thermosets, a simplified stability test was conducted, where the films were immersed in a vial containing different aqueous and organic solvents for 7 days (Appendix A). Although it was found that CHCl_3_ disintegrated all thermosets (not dissolved), as shown for other crosslinked networks containing eugenol and DVB, the samples exhibited better chemical stability toward DMSO compared to the thermosets with DVB and eugenol [55]. However, X(1,10-MVP–4T) was found to disintegrate when immersed in NaOH (1 M) and DMSO. Some discoloration was observed for all samples immersed in NaOH except for X(1,3-MVP–3T) and X(1,3-MVP–4T). The polarity of the thermosets was investigated through contact-angle measurements (CA) against water (Appendix A). This approach provided a better understanding of the hydrophobicity of the thermosets. All films possessed a hydrophobic character with CAs close to 78°.

In summary, the facile modification route of MVP resulted in DVB-like difunctional monomers with potential use in thermally crosslinked thiol-ene networks. However, electron-poor alkenes are also susceptible to self-initiation. We believe that self-initiation resulted in a less-defined network, which could explain the lack of clear DMA trends and gel content variations.

## 4. Conclusions

Lignin-derived ferulic acid (FA) can act as the basis for producing 2-methoxy-4-vinylphenol (MVP), resulting in a biobased versatile monomer platform. In this work, MVP was functionalized to prepare biobased monomers that could be readily used in solution and emulsion polymerization, thus expanding its use in free-radical polymerization targeting thermoplastic applications. It was found that the biobased homopolymers exhibited lower *T*_g_ values compared to other styrene-like polymers due to the increased mobility provided by the varying functional groups. Moreover, the MVP-platform monomers were successfully copolymerized with styrene, resulting in stable emulsions with a biobased content of up to 10 wt.%. However, future endeavors must focus on increasing the biobased content in PS emulsions and perhaps partially replacing PS with MVP.

Additionally, MVP was functionalized to prepare divinylbenzene (DVB)-like monomers that were thermally cured with thiol crosslinkers bearing three (3T) or four (4T) thiol units, in an attempt to investigate their potential in thermosetting applications. Their thermomechanical properties were influenced by both the length of the aliphatic chain connecting the MVP units and their crosslinking density (v_e_). In particular, *T*_g_ was tuned by increasing the aliphatic length, thus increasing for the samples containing 4T since their v_e_ values were greater compared to their 3T counterparts. Their mechanical properties were found to be mainly affected by v_e_, where a better performance above the glass transition was achieved for samples with higher v_e_. However, to unlock the full potential of MVP-derived thermosets, their preparation protocol has to be further fine-tuned, as competition between homopolymerization of the DVB-like monomers and crosslinking with thiol-bearing reagents was identified.

Through this work, we demonstrated the possibilities of MVP-derived monomers in thermoplastic and thermosetting applications. We created a library of monomers that can undergo radical polymerization, targeting new biobased functional materials as alternatives to their fossil-based counterparts.

## Data Availability

All characterization data will be uploaded if the reviewers and/or editors find them useful for the readers of Polymers.

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
