# Peer review of "2-Methoxy-4-Vinylphenol as a Biobased Monomer Precursor for Thermoplastics and Thermoset Polymers"

_polymers, 2023, doi:10.3390/polym15092168_

Round 1

Reviewer 1 Report

This research topic "2-Methoxy-4-vinylphenol as a bio-based monomer precursor for thermoplastics and thermoset polymers" is very interesting for the development of bio-based materials. The paper is well written and most of data seem technically sound. However, there are still doubts and improvement points to make the manuscript clear as follows:

1.     In Table 2, why are the Tg results derived from DSC so different from those observed in DMA? For example, the Tg values of X(1,10-MVP–3T) is 4.5oC based on DSC while 30oC according to DMA.

2.     The various physical parameters appearing in Eq. S4 need to be further explained clearly. Why the values of E' are fixed as  (at 80oC) for every samples?

3.     “4. Discussion” should be “4. conclusion”.

Author Response

  1. We thank the reviewer for the comment. It is well-known from the literature that Tgs obtained by different techniques may be different. That is due to different heating/cooling protocols used.
  2. We thank the reviewer for the comment. All of the parameters used in equation S4 are described in the SI of the respective section (part B, section 9). To use equation S4, we need to use E’-values at the steady state (plateau) after the glass transition region. Since some of the thermosets broke during the DMA testing at ca. 80 oC, as described in the main manuscript, the highest value we could use was 80 oC.
  3. We thank the reviewer for this comment. We have corrected this in the revised version of the manuscript.

Reviewer 2 Report

In this paper, the authors synthesize thermosets from bio-based styrene-like monomer called 2-methoxy-4vinylphenol (MVP). The paper is very well written and the authors have provided all the data from the characterization of the material. I am happy with the work and the presentation and suggest the paper for publication.

Author Response

We are very grateful for the positive comments and feedback from the reviewer.

Reviewer 3 Report

the draft tilted with "2-Methoxy-4-vinylphenol as a bio-based monomer precursor 2 for thermoplastics and thermoset polymers" presented interesting work, based on FA-derived styrene-like monomer, referred to as 2-methoxy-4-vinylphenol (MVP), was used as the platform to prepare functional monomers for radical polymerizations. Hydrophobic bio-based monomers derived from MVP were polymerized via solution and emulsion polymerization resulting in homo- and copolymers with a wide range of thermal properties, thus show-casing their potential in thermoplastic applications.

However, i recommend to be published after minor revision.

Some written mistakes  such as:  you forgot the parenthesis for the reference number 18,
also try to arranged the figures in the draft parallel with the supplementary file according there appearance in the draft  and so on , check the entire draft also, i recommend you to move the tables from the supplementary file into the original draft to be more useful.

the conclusion are missing!

Author Response

We thank the reviewer for valuable comments. We have corrected the erroneous reference. Unfortunately, we had made a mistake with the Conclusion-heading which has now been corrected.

We prefer not to chance the order of the Supplemental Information (SI) since we believe that it will the be less available to those readers that would like to use the experimental section as a stand-alone material. However, we do hope that we have made sure that the referencing to SI is clear so that it is easy to find the appropriate section in the SI.